# Investigating the Health Implications of Whey Protein Consumption: A Narrative Review of Risks, Adverse Effects, and Associated Health Issues

**DOI:** 10.3390/healthcare12020246

**Published:** 2024-01-18

**Authors:** Edda Cava, Elvira Padua, Diego Campaci, Marco Bernardi, Fares M. S. Muthanna, Massimiliano Caprio, Mauro Lombardo

**Affiliations:** 1Clinical Nutrition and Dietetics, San Camillo Forlanini Hospital, Rome, cir.ne Gianicolense 87, 00152 Rome, Italy; ecava@scamilloforlanini.rm.it; 2Department of Human Sciences and Promotion of the Quality of Life, San Raffaele Open University, Via di Val Cannuta, 247, 00166 Rome, Italy; elvira.padua@uniroma5.it (E.P.); diego.campaci@gmail.com (D.C.); marco.bernardi@studenti.uniroma5.it (M.B.); massimiliano.caprio@uniroma5.it (M.C.); 3Pharmacy Department, Faculty of Medicine and Health Sciences, University of Science and Technology-Aden, Alshaab Street, Enmaa City 22003, Yemen; f.mothana@aden.ust.edu; 4Laboratory of Cardiovascular Endocrinology, San Raffaele Research Institute, IRCCS San Raffaele Roma, Via di Val Cannuta, 247, 00166 Rome, Italy

**Keywords:** whey protein, health risks, collateral effects, liver damage, kidney damage, physical activity, acne, bone mass, sedentary individuals, chronic intake, review study

## Abstract

This narrative review critically examines the current research on the health implications of whey protein (WP) supplementation, with a focus on potential risks and adverse effects. WP, commonly consumed for muscle building and weight loss, has been associated with various health concerns. Our comprehensive analysis involved a thorough search of multiple databases, resulting in the inclusion of 21 preclinical and human studies that collectively offer a detailed overview of WP’s health impacts. The review reveals significant findings, such as WP’s potential link to liver and kidney damage, alterations in gut microbiota, increased acne incidence, impacts on bone mass, and emotional and behavioural changes. These findings underscore the complexity of WP’s effects on human health, indicating both beneficial and detrimental outcomes in relation to different posologies in a variety of settings. Our study suggests caution for the protein intake in situations of hepatic and renal compromised functions, as well as in acne susceptibility, while possible beneficial effects can be achieved for the intestinal microbiota, humoral and behavioural level, and finally bone and muscle mass in elderly. We emphasizes the importance of balanced WP consumption and call for more in-depth research to understand its long-term health effects. Health professionals and individuals considering WP supplementation should be aware of these potential risks and approach its use with informed caution.

## 1. Introduction

Whey protein (WP), a key component of milk proteins [1], has gained widespread popularity for its purported benefits in muscle building and weight management. However, its increasing consumption has raised concerns about potential health implications, which this review aims to investigate. WP characteristics can vary based on multiple factors such as method of casein precipitation, storage conditions, heat treatment, and other variables [2]. WP are commonly subjected to various processing methods, including ultra and/or microfiltration or ion exchange, resulting in the creation of WP isolate (containing 90–95% protein, minimal fat, lactose, and mineral content), WP hydrolyzed to achieve more readily absorption and less antigenic reactions, as well as WP concentrate (with protein content ranging from 20% to 85%, along with varying amounts of fat, lactose, and minerals) [3]. Table 1 provides a comprehensive overview of types and concentrations of WP. WP provides a rich essential amino acids (AAs) source being rich in both sulphur-containing and branched-chain AAs. Table 2 provides a detailed breakdown of the key constituents found in WP, including Beta-Lactoglobulin, Alpha-Lactoalbumin, and various Immunoglobulins, highlighting their concentration percentages.

WP supplementation has gained considerable attention in the field of health and sports. Athletes often use WP seeking to improve muscle mass, strength or body composition [4]. Nutritional and physiological properties of WP supplementation and its effects on body composition and performance are widely studied, but the results are not always consistent due to the lack of study standardization in terms of samples, tests used, duration, amount or types of protein supplements and more [5].

Numerous studies have extensively explored the effects of WP supplementation in sports. These investigations have led to the formation of broadly accepted recommendations regarding WP use in athletic contexts. The meta-analysis of Davies et al., showed that WP supplementation has a small to medium ergogenic effect on the recovery of muscle function after endurance training, however, less than half of the included studies reported an overall beneficial effect [6]. Notably, the timing and dosage of WP intake play a critical role in maximizing its benefits for athletes. Pre- and post-training supplementation has been shown to significantly impact muscle recovery and growth. For instance, Kim et al. [7] demonstrated that whey protein consumed immediately before or after an exercise session significantly enhances muscle protein synthesis (MPS). Moreover, the amount of WP intake is crucial, with studies suggesting an optimal dosage range to maximize muscle recovery and growth without adverse effects. Naclerio and Seijo [8] provides insights into the ideal quantity of WP intake for athletes to facilitate muscle repair and growth post-exercise. Other studies report improvements in muscle mass [9], also due to an increase in the recruitment of satellite cells [10], strength [9], performance [5] and improvement in recovery times and in body composition [5], particularly if the oral intake is combined with resistance training [11], while other results are not in agreement [12]. The key amino acid in stimulating MPS is leucine, and its presence in WP is likely a major factor contributing to WP’s effectiveness in stimulating MPS when compared to other protein supplements, such as soy or casein [13].

Little is known about the side effects and possible long-term detrimental consequences of WP supplementation. Doubts have been raised especially regarding the effects of chronic use, at high doses by sedentary subjects, on liver and kidney function [3,14]. However, these doubts are not shared by all the researchers, who have refuted the previous conclusions [15]. Other concerns have been raised about whether WP may elicit allergic responses [16] or symptoms of lactose intolerance. Over-supplementation with WP may contribute to the excess of animal protein in the diet, increasing the potential risk of health-related conditions such as type 2 diabetes (T2DM) [17]. The intake of WP, especially in excessive amounts, may influence the onset of T2DM through several physiological mechanisms. Firstly, the high content of branched-chain amino acids (BCAAs) in WP may lead to insulin resistance, a key factor in the development of T2DM. This is because BCAAs, especially leucine, can activate the mammalian target of rapamycin (mTOR) pathway, which plays a crucial role in insulin signaling and glucose homeostasis [18,19]. Chronic activation of this pathway is associated with impaired insulin signaling and reduced glucose uptake into muscle cells, contributing to hyperglycaemia. Furthermore, excessive protein intake can increase the pancreas’ demand for insulin, potentially leading to beta-cell dysfunction over time [20,21].

Furthermore, the rapid digestion and absorption of WP leads to a rapid and significant increase in blood amino acid and insulin levels, potentially desensitising insulin receptors and contributing to insulin resistance [22]. Moreover, during the COVID-19 pandemic, significant changes occurred in physical activity [23] and dietary habits, especially in the types of protein sources consumed [24]. These potential health concerns, amid conflicting research perspectives, underscore the need for a comprehensive review of WP’s health implications.

Through a critical evaluation of existing scientific literature, this review aims to provide a balanced perspective on WP’s health risks and benefits.

This table outlines the various types of Whey Protein (WP) supplements, highlighting their respective protein concentrations. It also details the main protein components found in whey, along with their relative concentration percentages. The main proteins in whey and their concentration (relative to the total whey proteins) are β-lactoglobulin (~55%), αlactalbumin (~20%), blood serum albumin (~7%), immunoglobulins (~13%) and minor proteins (~5%).

This table provides a detailed breakdown of the key constituents found in Whey Protein (WP), along with their respective concentration percentages. The table covers a range of components from major proteins like Beta-Lactoglobulin and Alpha-Lactoalbumin, to less abundant but significant elements such as various Immunoglobulins, Lactoferrin, and enzymes.

## 2. Materials and Methods

The review was conducted using multiple databases including PubMed, Scopus, and Google Scholar, and adheres to the Scale for the Assessment of Narrative Review Articles (SANRA) guidelines [25]. The following search terms were used: “whey protein” OR “whey protein supplementation” AND “adverse effects” OR “side effects” OR “liver toxicity” OR “kidney function” OR “acne” OR “gut microbiota” OR “mood swings” OR “anger” OR “allergic reactions” OR “digestive issues” OR “contraindications”. This strategic approach aimed to yield a comprehensive set of studies relevant to the adverse effects of WP supplementation.

### 2.1. Inclusion Criteria

Included in the review were clinical trials, preclinical studies, observational or case-control studies, case-series and prospective cohort studies. The studies were published between 1 January 1990, and 31 October 2023, and were in English.

### 2.2. Exclusion Criteria

The exclusion criteria were (1) review articles, conference abstracts, commentaries, and (single) case reports, and (2) duplicate publications emanating from the same trials. Additionally, any review featuring WP supplementation side effects in the title or abstract was scrutinized to extract further relevant references. Notably, studies reporting positive effects of WP were omitted, as they were considered outside the intended scope of this review. The decision to exclude studies with non-significant results or positive effects of WP was made to focus this review on potential adverse effects. However, it is acknowledged that this exclusion might limit the breadth of our analysis and potentially introduce a selection bias. While these studies could offer a more balanced view of WP’s overall health impact, their exclusion was necessary to adhere strictly to the review’s predefined scope.

### 2.3. Data Extraction

A flow diagram illustrating the selection procedure is presented in Figure 1.

This figure illustrates the approach taken in the literature review process, from initial database search to the final selection of studies included in the review. It details the number of papers identified, screened, and evaluated, as well as the reasons for excluding certain studies, ultimately leading to the final selection of relevant studies for inclusion in the review.

Initially, 14,274 papers were identified through database searches. Upon screening, 11,419 records remained after removing duplicates. Studies were deemed irrelevant if they did not directly address the adverse effects of WP supplementation or if they focused on populations not within the scope of our review, such as pediatric or geriatric subjects. Indeed, infant formula are included in the discussion but not within results, as WP in this case would not be considered a supplement to regular diet nutrients, but a main constituent of the nutritional intake.

Incorrect article types included those that were purely theoretical without empirical data, or were editorial opinions rather than research studies. The full text of 176 studies was evaluated. From these, 148 were excluded after reviewing the titles and abstracts for reasons such as irrelevance to the research question, incorrect article type, and non-conforming study populations. Following a more detailed examination, 7 articles were excluded due to lack of relevant data (n = 3), inadequate methodologies (n = 2), non-significant results (n = 1), and being outside the scope of the review (n = 1). Ultimately, 21 studies met the inclusion criteria and were included in the review.

Two authors (E.C., and D.C.) independently extracted the required information from the selected studies. This included the first author’s name, publication year, trial participant characteristics (e.g., number of participants in intervention and control groups, mean age and age range), duration of the intervention, trial design, and definition of the intervention and related control. In instances of discrepancies between the two primary authors, a third author (M.L.) was consulted to reach a consensus.

## 3. Results

The review of 21 studies [26,27,28,29,30,31,32,33,34,35,36,37,38,39,40,41,42,43,44,45,46] revealed diverse outcomes concerning the impact of WP on various health aspects. The research identified twelve human studies, eight pre-clinical studies, and one using an intestinal simulator. Details of these studies on health and side effects from WP supplementation are presented in Table 3 for human studies and Table 4 for preclinical studies.

### 3.1. Liver Function

The impact of WP on liver function appears to vary across studies. In human research, Chitapanarux et al. [26] observed beneficial effects, including a decrease in hepatic steatosis and oxidative stress in NASH patients. Meanwhile, Schlickmann et al. [36] reported slight alterations in liver function markers, such as aspartate aminotransferase (AST) and urea levels, in gym users. Thus, Nhean et al. [37] demonstrated that 8% of patients using appearance- and performance-enhancing supplements (APES), mainly represented by WP, experienced grade 3–4 ALT/AST elevations. In preclinical studies, Deminice et al. [43] found increased hepatic oxidative stress markers in rats supplemented with WP, and Gürgen et al. [44] reported liver toxicity and increased inflammatory markers in untrained mice following WP supplementation. Similarly, Nunes et al. [42] found elevated liver/kidney toxicity markers in sedentary WP mice, but noted that combining supplements with resistance training mitigated these effects.

### 3.2. Kidney Function

Studies on kidney function following WP consumption have mixed results. In human studies, Bauer et al. [35] observed an improvement in estimated glomerular filtration rate among sarcopenic older adults consuming WP-micronutrient drinks. Hattori et al. [33] identified increased urinary calcium and decreased urinary pH in individuals consuming WP and Nhean et al. [37] found elevated serum creatinine levels in 12% users of APES who were under a concurrent chronic antiviral therapy. Aparicio et al. [40] in a study with sedentary mice, reported increased renal volume and calcium excretion with high protein intake, though these effects were less pronounced with endurance exercise. Confirming the data of the already cited study by Nunes et al. [42], Amanzadeh et al. [38] demonstrated that increased WP intake led to decreased urinary pH, and increased ammonia and calcium concentrations in the urine of murine models.

### 3.3. Acne

Research has highlighted a notable link exploring the connection between WP consumption and acne development. In human studies, Simonart et al. [29], Silverberg et al. [30], Pontes et al. [31], and Cengiz et al. [32] consistently found an increase in acne severity and the development of lesions among individuals consuming WP supplements. Improvements were experienced after discontinuation of WP supplements. The consistency in findings across these studies indicates a strong link between WP supplementation and increased acne severity, particularly pronounced in specific groups such as male bodybuilders and adolescents.

### 3.4. Gut Function and Microbiota

Several papers suggest that WP can influence gut microbiota composition and function, although the effects may vary depending on individual gut microbiomes and the specific context of consumption. In human research, Moreno-pérez et al. [34] observed changes in the intestinal microbiota of cross-country runners, including an increase in the Bacteroidetes phylum and a decrease in beneficial bacteria such as Roseburia and Blautia. Thus Zhou et al. [46], found that donkey WP and hydrolysate improved gut microbiota and physiological functions in aging mice, enhancing weight gain and liver health, and reducing oxidative stress markers. Additionally, Sanchez-Moya et al. [45], in their in vitro study, reported an increase in beneficial bacteria like Bifidobacterium and Lactobacillus, as well as an enhanced production of short-chain fatty acids, which are crucial for gut health.

### 3.5. Emotional and Behavioural Influences

Emerging evidence suggests that WP may affect emotional and behavioral responses. In a human study, Santos et al. [28] identified a correlation between higher protein intake and increased expression of anger among male bodybuilders. Orosco et al. [39] studied murine models and noted increased serotonin production and reduced anxiety after the consumption of alpha-lactalbumin, a component of WP. The plasma ratio of tryptophan over the sum of its competitor large neutral amino acids (Trp/LNAAs) that may represents a non-pharmacologic measure of reward accounting for the hedonic state of the animals and may produce behavioural effects known by involving enhanced serotonin synthesis and transmission, with antidepressant-like action.

### 3.6. Bone Metabolism

Studies investigating the impact of WP on bone metabolism have produced diverse results. In a pre-clinical study, Amanzadeh et al. [38] found that high protein intake in murine models led to decreased bone density along with changes in urinary pH and calcium concentration. In human research, Hattori et al. [33] observed increased urinary calcium excretion and decreased urinary pH in individuals consuming WP. Bauer et al. [35] investigated the impact of WP consumption in older adults. Their study reported modifications in serum calcidiol and calcium levels, along with a decrease in parathyroid hormone levels. However, contrasting these findings, Zhu et al. [27] reported no significant impact on bone mineral density from WP consumption on menopausal women. These varied findings in bone metabolism studies underscore the complex nature of WP’s effects, suggesting that individual factors may significantly influence outcomes.

### 3.7. Other Health Effects

In a pre-clinical setting, Delamaire et al. [41] observed in rats significant increase in weight gain, food intake, and several metabolic parameters including serum insulin, leptin, and triglycerides. Additionally, this study noted changes in pancreatic β-cell number and adipocyte size, suggesting broad metabolic effects of WP in animal models.

Overall, the collected data from these studies presents a multifaceted view of WP’s health effects, highlighting both potential risks and areas where further research is required to draw definitive conclusions.

## 4. Discussion

The world of fitness and gyms is invaded by products, with high percentages of athletes declaring that they regularly take protein supplements often without any advice from a specialist [5,9,10,11]. The consumption of protein supplements grows proportionally to the level of sports practice (higher consumption in elite athletes) and to the volume of weekly training, with higher numbers in individual sports compared to team sports [13]. In the field of athletic performance, WP is highly valued for its effectiveness in improving muscle recovery and overall physical endurance. Athletes, who represent a significant proportion of WP consumers, often rely on its high biological value and essential amino acid profile, particularly its leucine content, to support muscle synthesis and recovery after exercise. WP’s role in accelerating recovery processes, reducing muscle soreness and potentially increasing muscle strength and size has been documented in numerous studies [47,48]. Furthermore, the ability of WP to help optimize body composition, which is crucial for athletes involved in weight-sensitive sports, is another aspect of its widespread use [49,50]. However, it is crucial to emphasise the importance of appropriate dosage and timing of WP intake, as improper use can lead to adverse effects. Tailored nutritional strategies are therefore recommended for athletes using WP supplements, taking into account individual athletic needs and health status.

### 4.1. Liver Function

The liver plays a fundamental role in protein metabolism, being the main receptor of amino acids introduced exogenously and produced endogenously by our body. The liver can use amino acids for energy purposes (gluconeogenesis) or to synthesize new proteins. The byproduct of degradation of the amino acids used for energy purposes in the liver is a substance called urea, which is then filtered by kidneys and later excreted. Although some studies argue that increased protein intake can lead to long-term damage to the liver function, few evidence are available in human studies, such as Nhean et al. who found elevation of liver enzymes and creatinine blood levels in subjects under chronic drug therapy [37], and Nunes et al. (2013) who showed alteration of hepatic and renal function caused by protein supplementation (higher levels of ALT and AST) being more evident in sedentary conditions, while resistance training seems to have a sort of buffer effect limiting its consequences [42]. The protective action of physical activity against liver damage seems in accordance with preclinical studies on rats, indeed Gurgen et al. demonstrated that long-term protein supplementation led to elevated levels of liver toxicity, apoptotic signals and inflammation (increased production of IL-6, IL-7, TNF-alpha) in untrained rats [44]. Moreover, a study by Deminice et al. (2015) with protein supplementation showed higher levels of hepatic oxidative stress comparing a group of rats not supplemented [43]. The hypothesis is that in the presence of physical exercise, more amino acids are brought to the muscles for protein synthesis, with a smaller amount being consequently taken to the liver [36]. On the contrary, some studies support the beneficial action of WP on the liver, especially in the case of a pre-existing condition. Chitapanarux et al. (2009) found that protein supplementation was associated with an improvement in the pathological condition and liver oxidative stress in the presence of NASH [26]. These conflicting results suggest that while WP can be beneficial in certain conditions, its indiscriminate use without guidance or specific indications, particularly in sedentary individuals, may pose risks to liver health.

### 4.2. Kidney Function

The degradation of AA leads to the production of nitrogenous waste, which must be expelled by kidneys through the production and excretion of urea. The greater the protein intake in the diet, the greater the urea production to eliminate excess nitrogen, with a consequent higher filtration activity at the renal level [47]. According to Aparicio et al. [40], a chronically protracted high-protein diet (associated with WP supplementation) leads to increased renal weight and urinary calcium excretion, which could be preliminary signs of impaired renal function and damage. The effects also seem to be more marked when combined with a sedentary lifestyle [38]. Alterations in AST concentrations and in urea production, which are alarms of possible impaired renal and liver function, seem to be particularly widespread within the world of gyms among those claiming to make frequent use of protein supplements [36]. However, some authors argue that the increase in kidney activity caused by a high protein regimen can also be seen as a simple adaptive response, not necessarily associated with kidney damage. There is a lack of scientific evidence supporting that a high-protein diet can lead to chronic kidney disease in healthy subjects [42]. Particular attention should be paid to the use of WP in subjects with nephrolithiasis, since some studies show that a high protein regimen can be associated with variations in lithogenic parameters (such as urinary calcium) with a tendency to the formation of kidney stones [33]. To date, the nutritional guidelines recommend reducing the intake of proteins, especially animal proteins, in case of kidney stones diagnosis [51,52].

Another point of attention should be the case of subjects undergoing chronic drug therapies that could interfere with renal or liver function. Indeed, among 50 participants under HIV pre-exposure prophylaxis with daily tenofovir disoproxil fumarate/emtricitabine, being part of a HIV high risk community, 12% (Three) users of supplements, mainly WP and creatine or anabolic steroids, had elevated serum creatinine, while 8% (two) users experienced grade 3–4 ALT/AST elevations versus not users subjects [37]. The implications of these findings are crucial, highlighting the need for careful consideration of WP supplementation, especially in individuals with pre-existing renal conditions or those on chronic medications.

### 4.3. Acne

Acne is caused by an overproduction of sebum by the sebaceous glands, which leads to the formation of papules and pustules on the skin, particularly in the facial area. In addition to genetic and hormonal factors, nutrition represents a determining factor in its onset, especially the consumption of milk and dairy products, including WP. In fact, these foods cause an increase in IGF-1 production, with consequent cell proliferation in the skin, hyperproduction of sebum and the onset of acne [29]. A lower consumption of milk and insulinotropic dairy products can lead to a lower incidence of this pathology [30]. Nutrition not only plays a key role in the onset of acne, but is also important in preventing its aggravation. In particular, fatty foods such as dairy products make it worse, while fruit and vegetables help improve it [31]. High glycemic index and load foods represent also a contributing factor, along with gender and ethnicity [32]. A study by Pontes et al. (2013) conducted among gym goers demonstrated how the incidence and severity of acne increase as the use of protein supplements continues over time [31]. The same phenomenon seems to emerge from the reports of Simonart et al. (2012) conducted on 5 healthy adults who developed acne after starting WP use [29]. The results overlap with those highlighted by another report by Cengiz et al. (2017) [32]. It also seems that suspending protein supplementation is necessary in order to have a significant improvement in the pathological picture, while the pharmacological treatment alone results insufficient [38]. Close attention should also be paid to the problem of prohibited substances, such as anabolic steroids, which can often be hidden in common protein supplements and be taken without consumer’s knowledge [53]. Androgens are in fact among the main responsible for the hypertrophy of the sebaceous glands and the consequent production of sebum [54]. Given the strong association between WP and acne, particularly in specific demographics, this warrants further research to establish definitive causal relationships.

### 4.4. Gut Function and Microbiota

According to a study by Moreno-Perez et al. (2018), WP supplementation, while not causing changes in the pH and ammonia content of the feces, can cause an imbalance in bacterial concentrations with an increase in the bacteroidetes phylum, which is potentially negative for the microbiota as it can be associated with increased inflammation and obesity [34]. Meddah et al. (2001) have, however, experimentally found a possible positive action of WP in the intestine, as it reduces the concentration of bacteroides fragilis and increases the production of short-chain fatty acids [55]. These results were confirmed by Zhou et al. (2022) [46], according to whom protein supplementation is associated with an improvement in intestinal bacterial concentrations (increase of Lactobacillus and reduction of Helicobacter), reduction of oxidative stress and potential slowing down of the aging processes. Sanchez-Moya et al. (2017) confirm the prebiotic and beneficial effect on the intestinal microbiota, given by increased concentrations of Bifidobacterium and Lactobacillus and increased production of short-chain fatty acids, also supporting their possible action on obesity prevention [45]. Furthermore, WP seem to help the correct development of the infant microbiota as well as prevent the onset of escherichia coli [56]. Similarly, pea proteins like WP also seem to play a significant beneficial action on the microbiota in terms of bacterial species concentration and short-chain fatty acid production [57]. These studies indicate a complex interplay between WP supplementation and gut health, suggesting both beneficial and potentially adverse effects depending on individual conditions.

### 4.5. Emotional and Behavioural Influences

The amino acid tryptophan, abundantly contained in proteins of animal origin, is the main precursor for the synthesis of serotonin, a neurotransmitter involved in mood regulation. Eating habits are able to influence the production of serotonin, and consequently our mood. Meals rich in carbohydrates, especially those with a high glycemic index, strongly stimulate its production [58]. Serotonin affects mood by reducing anxiety, aggression and depression [59]. Some authors argue that the high intake of proteins, specifically of animal origin or WP, determines a greater availability of BCAAs contained in them; these seem to be able to hinder tryptophan in the passage of the blood-brain barrier, reducing its availability and therefore the production of serotonin. In fact, it seems that in bodybuilders, typically known to follow a high-protein diet and often a concomitant use of protein supplements, the expression of anger is higher than in the normal population [58]. On the contrary, some authors have shown that meals with a high leucine content (abundantly found in WP) are able to reduce the perception of tiredness and the expression of anger, as well as improve recovery and performance [59]. WP are also characterized by a high content of alpha lactalbumin, rich in tryptophan, the contribution of which seems to be related to a greater availability of this amino acid with consequent greater synthesis of serotonin and beneficial effects on mood. This phenomenon was confirmed in a study by Orosco et al. (2004), who demonstrated how the administration of alpha lactalbumin to experimental mice is associated with anxiolytic and rewarding effects [39]. The increased production of serotonin determined by the intake of alpha lactalbumin also leads to an improvement in the quality of sleep, with a consequent benefit in terms of mood and cognitive performance the following day [60]. Markus et al. (2002, 2000) have also demonstrated its ability to reduce perception of stress, diminish depressive feelings and lead to better cognitive performances under stress [61,62]. Recent studies have also revealed the positive action of milk and dairy products in slowing down cognitive decline in old age and reducing the incidence of pathologies such as Alzheimer’s disease [63]. A work by Kita et al. (2018) showed that milk and WP can correlate with a significant improvement in memory and attention in adult subjects [64]. The apparent impact of WP on mood and behavior points to the need for more studies about how protein supplementation interacts with psychological and physiological factors.

### 4.6. Bone Metabolism

Currently osteoporosis is a pathology of particular interest, due to the worldwide ageing of the population, with its numbers constantly increasing [65]. In the past it was thought that high-protein diets (especially with proteins of animal origin) and a lack of fruit and vegetables were related to a greater mobilization of calcium from bone tissue and consequent hypercalciuria, predisposing to a greater probability of osteoporosis [66]. Recent studies have reconsidered this assumption, questioning the existence of a correlation between meat consumption and increased calcium excretion [67]. On the contrary, it emerged how the intake of proteins of plant or animal origin is essential for the maintenance of muscle mass and for promoting intestinal calcium absorption in the elderly [68]. Similarly, as demonstrated by Zhu et al. (2011), even the long-term use of WP as protein supplementation in menopausal women does not seem to correlate with a more rapid process of worsening of skeletal mass and strength [27]. Similar results were also obtained on laboratory rats [40]. In addition to this, it has been confirmed that a protein intake higher than the common guidelines and the use of WP in the elderly are particularly useful for the prevention of sarcopenia and for the maintenance of muscle mass, essential for reducing the risk of falling [69]. In these patients, monitoring of serum vitamin D levels is essential, particularly in elderly subjects with chronic musculoskeletal pain [70]. These findings contribute to a growing body of evidence that challenges previous assumptions about protein intake and bone health, particularly in aging populations.

### 4.7. Whey Protein in Infant Formula

In March 2023 FDA has updated its claim guidance for the labeling of infant formula, first issued in September 2016, requiring a statement weather infant formula contains enzymatically hydrolysed WP isolate, coming from cow’s milk, since these formula should not be considered hypoallergenic and fed to infants allergic to milk [71]. Little to very little evidence can support a qualified health claim for partially hydrolyzed WP infant formula and a reduced risk of atopic dermatitis in partially breastfed and exclusively formula-fed infants [72]. Another potential field of risk with hydrolysed WP in infant formula, is suggested by a report describing infants with cholestasis receiving this particular formula with increased risk of Vitamin K Deficiency (RR 25.0 [6.4–97.2], *p* < 0.001) compared with infants receiving regular formula. To the best of our knowledge this is the first and only record of risks related to WP intake in infant formula, other than gastrointestinal tolerance [73]. These concerns emphasize the importance of rigorous testing and clear labeling of infant formulas to ensure safety and suitability for vulnerable populations.

### 4.8. Miscellaneous Fields for Future Research on WP

In the end, very little evidence is available on WP byproducts that can be generated before and after ingestion. Notably, WP can incur in glycation by interaction with reducing sugars and carbohydrates, via Maillard reaction, depending on the preparation process and heat treatments. Modification of their functional and sensory properties result in variable increase in solubility, thermal stability, antioxidant activity, and emulsification or foam stabilization [74].

The nutritional and potential health effects of WP-polysaccharides conjugates is still under research, with reports suggesting that these WPI-glucose conjugates may promote prostate cancer cells proliferation in different ways [75,76], through stimulation of cytokines production by macrophages or through immunomodulatory effects [76].

At last, interactions of WP with metal ions are known but not fully understood and explored in human nutrition and health. Compounds such as metalloproteins, metallocomplexes, nanoparticles, or aggregates may have potential biologically activity but unknown consequences. Although WP–metal interactions is more common in milk products with stainless steel surfaces during storage, transport, and processing, it is worth to mention that it can occur also during the process of purification of WP by magnetic metal nanoparticles. To date, no safety issues or toxic effects have been reported about it [77].

## 5. Conclusions

In conclusion, this review suggests that the adverse role of WP on our health is still to be clarified. This review underscores the multifaceted nature of WP supplementation, calling for a balanced perspective that considers both its potential benefits and its risks. Hepatic and renal functions are certainly affected by a greater protein intake, even if it is not yet clear whether this mechanism can lead to concrete damage and what is the upper limit dose. As far as acne is concerned, further studies are needed, as the available literature is currently very scarce and mostly limited to individual case studies. WP seem to have a beneficial effect on the composition of the intestinal microbiota and its functionality, as well as on a humoral and behavioural level, without any significant correlation with increases in anger levels. Finally, evidence does not clearly suggest a more rapid deterioration of bone mass in the elderly, but actually seems in favor of the maintenance of good muscle mass. Table 5 summarises the results of the review and provides takeaway messages. Further and more in-depth studies on the possible detrimental effects of WP supplementation are warranted.

## Figures and Tables

**Figure 1 healthcare-12-00246-f001:**
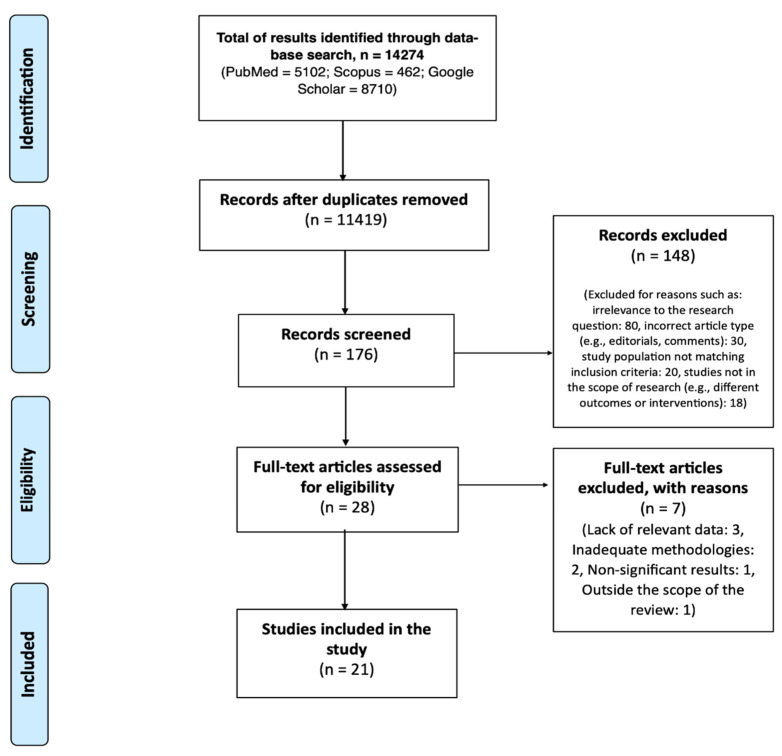
Flow Diagram of the Literature Search and Study Selection Process.

**Table 1 healthcare-12-00246-t001:** Types and Concentrations of Whey Protein Supplements.

WHEY PROTEIN (WP) Type	Concentration
WP Isolate	90–95%
WP Concentrated	25–99%
Hydrolysed WP	Variable
Undenatured WP	Variable, common range 25–99%

**Table 2 healthcare-12-00246-t002:** Composition and Concentration of Whey Protein Constituents.

WHEY PROTEIN Constituents	% Concentration
Beta Lactoglobulin (β-LG)	50–55
Alfa-Lactoalbumin (α-LA)	15–20
Immunoglobulin A (IgA)	<15
Immunoglobulin G1 (IgG1)	
Immunoglobulin G2 (IgG2)	
Immunoglobulin M (IgM)	
Bovin Serum Albumin (BSA)	5–10
LActoferrin (Lf)	<1–2%
Lysozyme (Ly)	<1%
Lactoperoxidase (Lp)	<1%
Casein Macropeptides	<10%
Sulphydryl oxidase	<1%
Superoxide Dysmutase	<1%

**Table 3 healthcare-12-00246-t003:** Human Health and Side Effects from Whey Supplementation: A Summary.

First Author	Year	Study Design	Participants	N. of Patients	Age	Dose (per day)	Follow-Up Period	Outcomes	Ref.
Chitapanarux	2009	open labeled pilot study	Male and female with NASH	38	15–60	20 g WP	12 weeks	↓ Hepatic steatosis ↓ Oxidative stress ↑ GSH	[26]
Zhu	2011	RCT	Healthy menopausal women	210	70–80	30 g of WP vs. 2.1 g of protein	2 years	↔ hip BMD ↔ femoral neck strength ↑ IGF-1 in WP group	[27]
Santos	2011	Cross-sectional study	Male bodybuilders	127	17–44	Not specified, but included regular diet and supplements	N/A	↑ Anger scores and anger expression above average ↓ Anger management and inward anger below average Association between ↑ weekly protein intake and ↑ anger expression	[28]
Simonart	2012	Case series	Healthy male adult bodybuilders	5	19–35	Not specified	6 months	↑ Development of moderate to severe acne after whey protein consumption ↑ Acne severity according to GAGS ↔ No change with standard acne treatments for some; complete clearance for one after discontinuing whey protein and initiating topical treatment ↔ Partial regression with topical treatments for others	[29]
Silverberg	2012	Case series	Athletes and adolescents	5	14–18	Not specified	N/A	↑ cystic acne, papules, pustules, and a few comedones on bilateral cheeks WP discontinuation led to acne improvement	[30]
Pontes	2013	prospective observational study	Men and women	30	18–30	Not specified	60 days	↑ Comedones, papules, pustules counts over time ↔ No significant change in scar count ↑ Severity of acne according to Leeds Acne Grading System over time ↔ No influence of sex or family history on lesion increase	[31]
Cengiz	2017	Retrospective analysis of case series	Male adolescent users of protein-calorie supplements	6	16–18	Not specified	3 months	↑ Acne lesions post protein supplement initiation Acne localization to the trunk, sparing the face ↓ Acne severity with discontinuation of supplements and treatment ↔ No other health anomalies detected in blood tests	[32]
Hattori	2017	Interventional study	Men and women	18	21–38	27 g WP vs. Albumin	3 days with a 1-week washout period between supplements	↑ mean protein equivalent of nitrogen appearance ↔ in lithogenic parameters ↑ Urinary calcium in 39% of subjects ↓ Urinary pH in 44% of subjects ↔ other urinary elements	[33]
Moreno- Pérez	2018	Randomized pilot study	Cross-country runners	24	18–45	1.8–3 g/kg	Ten weeks.	↔ Fecal pH ↔ Fecal water content ↔ Fecal ammonia ↔ Fecal SCFA concentrations ↔ Plasma malondialdehyde levels ↑ Bacteroidetes phylum ↓ Roseburia ↓ Blautia ↓ Bifidobacterium longu	[34]
Bauer	2020	RCT	Sarcopenic older adults	380	>65	21 g protein, 3 g leucine, 10 µg vitD and 500 mg calcium per serving	26 weeks in total (13-week RCT followed by 13-week OLE)	↑ eGFR in test group during RCT; no change during OLE Plateau of serum calcidiol and calcium levels after 13 weeks ↓ PTH levels in the test group during RCT and in former control groups during OLE Overall good tolerability of the WP-MND over the 6-month intervention period	[35]
Schlickmann	2021	Cross-sectional study	Gym users	594	37	Not specified	N/A	↑ Slight alterations in AST ↑ Slight alterations in urea levels	[36]
Nhean	2023	prospective, cross-sectional survey with retrospective, observational cohorts	High- risk HIV subjects consuming APES while under PrEP	50	>18 (median 32)	survey on regular usage	34	↑ grade 3–4 ALT/AST elevations ↑ serum creatinine	[37]

This table summarizes key findings from human studies on health and side effects resulting from whey protein supplementation. Details include first author, year, study design, participant characteristics, number of patients, age range, dosage, follow-up period, outcomes, and references. Abbreviations: 1,25D: 1,25-dihydroxyvitamin D, APES: Appearance- and performance-enhancing supplements, BMD: Bone Mineral Density, eGFR: Estimated Glomerular Filtration Rate, GAGS: Global Acne Grading System, GSH: Glutathione, Hepatic steatosis: Fatty liver condition, IGF-1: Insulin-like Growth Factor 1, Lithogenic parameters: Factors affecting stone formation in the urinary system, NASH: Non-alcoholic steatohepatitis, OLE: Open-Label Extension, PrEP: HIV pre-exposure prophylaxis, PTH: Parathyroid Hormone, RCT: Randomized Controlled Trial, SCFAs: Short-Chain Fatty Acids, WP: Whey Protein, WP-MND: Whey Protein Medical Nutrition Drink, ↑: Increase, ↓: Decrease, ↔: No change.

**Table 4 healthcare-12-00246-t004:** Preclinical Studies on Health and Side Effects from Whey Supplementation.

First Author	Year	Model	WP Dose	Follow-Up Period	Outcome	Ref.
Amanzadeh	2003	Male Sprague Dawley rats, (8 weeks old, Weight 232 g)	3.34 g/kg/day:	59 days	↓ urinary pH, ↑ ammonia and calcium concentration in urine in the ↑ protein group ↓ bone density at the femur level in the ↑ protein group	[38]
Orosco	2004	Male Wistar rats (Weight 200–250 g)	190 g/Kg alpha-lactalbumin	6 days	↑ Serotonin production ↓ Anxiety after alpha-lactalbumin consumption	[39]
Aparicio	2011	Albino male Wistar rats (Young, Weight 150 g) undergoing endurance training	Not specified	3 months	↑ Renal volume; ↑ Calcium excretion with high protein intake ↓ Effects of high protein with endurance exercise	[40]
Delamaire	2012	Male Sprague Dawley Rats	8.7–13 g/dL	15 days	↔ Presence of mesenteric fat ↓ Low neonatal weight ↑ Weight gain in puberty/adulthood ↑ Food intake ↑ Serum insulin, leptin, triglycerides ↑ Pancreatic β-cell number; ↑ Adipocyte size	[41]
Nunes	2013	Sedentary Wistar rats (250 to 300 g; 90 days old) undergoing endurance training	1.8 g/kg post-training	8 weeks	↑ Plasma ALT/AST; ↑ Liver/kidney toxicity with protein supplements, no training ↓ Liver/kidney toxicity with protein + resistance training	[42]
Deminice	2015	Wistar rats (weight 120 g)	Not specified	4 weeks	↑ Hepatic oxidative stress markers in protein-supplemented vs. control	[43]
Gürgen	2015	Young male Wistar albino rats (Weight 170 g) Untrained mice	Not specified	5 days and 4 weeks	↑ Inflammatory interleukins/TNF-alpha with 4 weeks protein supplementation ↑ Liver toxicity; ↑ Apoptotic signals	[44]
Sanchez-Moya	2017	In vitro (donor faeces)	Variable	24 h	↑ Bifidobacterium and Lactobacillus with whey supplementation. ↑ Production of short-chain fatty acids.	[45]
Zhou	2022	Female Kunming mice (age 6 weeks, weight 25–30 g)	Variable	7 weeks	↑ SOD concentrations; ↓ Oxidative stress with protein supplementation. ↑ Lactobacillus; ↓ Helicobacter; positive intestinal effects.	[46]

This table presents a summary of preclinical studies investigating the health impacts and side effects of whey protein supplementation. It includes information about the first author, publication year, model used in the study, whey protein dose, follow-up period, observed outcomes, and references. Abbreviations: ↑: Increased, ↓: Decreased, ↔: No change/Stable, ALT: Alanine Aminotransferase, AST: Aspartate Aminotransferase, SOD: Superoxide Dismutase, TNF-alpha: Tumor Necrosis Factor-alpha.

**Table 5 healthcare-12-00246-t005:** Summary of Key Findings and Conclusions on Whey Protein’s Impact on Various Health Aspects. This table shows an overview of the main research findings and takeaway messages on the effects of whey protein on liver function, kidney function, acne, gut function and microbiota, emotional and behavioral influences, and bone metabolism.

Topic	Key Findings	Conclusion/Takeaway Message
Liver Function	Moderate WP intake does not harm liver function in healthy individuals. Caution advised for pre-existing liver conditions.	Generally safe for liver, but caution needed for liver diseases.
Kidney Function	In individuals without kidney disease, WP intake within recommended limits is safe. Excessive protein can stress diseased kidneys.	Safe within limits for healthy kidneys, but caution needed for renal disease.
Acne	Mixed results, possible link in susceptible individuals.	More research needed to clarify the relationship.
Gut Function and Microbiota	Beneficial impact on gut health and microbiota, enhancing gut barrier and reducing inflammation.	Promising role in gut health, further research warranted.
Emotional and Behavioral Influences	No conclusive link to significant emotional or behavioral changes.	No primary association, individual responses may vary.
Bone Metabolism	Positive impact on bone health, potentially aiding in bone density and strength.	Promising role in bone metabolism, further research warranted.

## Data Availability

Not applicable.

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
