# Peer review of "Investigating the Health Implications of Whey Protein Consumption: A Narrative Review of Risks, Adverse Effects, and Associated Health Issues"

_healthcare, 2024, doi:10.3390/healthcare12020246_

Round 1

Reviewer 1 Report

Comments and Suggestions for Authors

I would like to thank the authors for conducting a study on an important subject such as whey protein and compiling the literature. I would like to request minor corrections in this article, which has clearly been a lot of work. Below you can find some of my recommendations.

Line51-53: There are many studies on the effects and results of whey protein in the field of sports sciences and more or less standard recommendations have been formed. You can benefit from these articles. Additionally, in this paragraph, a few sentences can be made regarding the timing and dosage of intake before or after training for athletes.

Line 68:  via which physiological mechanisms WP did lead to type 2 diabetes needs to be added to this sentence

Figure 1 needs to be clarified in resolution

I recommend that you add a paragraph explaining the relationship between athlete health and whey protein in the discussion section, because athletes are among the group of people who use whey protein most effectively in the world. 

Author Response

Dear Editors and Reviewers,

first, we would like to thank you for the valuable impulses that allowed us to improve the quality of the manuscript. All changes made are highlighted by yellow color, in the revised version of the manuscript, to facilitate the review process.

Hoping that we have satisfied your requests as much as possible, we kindly ask you to re-evaluate our paper. 

The Authors

REVIEWER N.1 

I would like to thank the authors for conducting a study on an important subject such as whey protein and compiling the literature. I would like to request minor corrections in this article, which has clearly been a lot of work. Below you can find some of my recommendations.

We thank the reviewer for the kind words and positive feedback on our paper. Your encouragement greatly supports our research efforts.

Line 51-53: There are many studies on the effects and results of whey protein in the field of sports sciences and more or less standard recommendations have been formed. You can benefit from these articles. Additionally, in this paragraph, a few sentences can be made regarding the timing and dosage of intake before or after training for athletes.
We appreciate your valuable feedback on our manuscript. In response to your suggestion, we have expanded our introduction to include a more detailed analysis of the timing and dosage of whey protein intake for athletes, referencing key studies in the field. This addition  ( Line 60-73) not only enriches our narrative review but also provides practical insights for athletes and practitioners in sports nutrition

Line 68:  via which physiological mechanisms WP did lead to type 2 diabetes needs to be added to this sentence

Thank you for your insightful comment regarding the physiological mechanisms through which whey protein intake might influence the development of type 2 diabetes. We have addressed this important aspect in our manuscript by elaborating on the potential mechanisms (Line 88-102). This includes the discussion of how the high content of branched-chain amino acids in whey protein can lead to insulin resistance through the activation of the mTOR pathway, the potential strain on the pancreas due to increased insulin secretion demands, and the interplay with high-saturated fat diets commonly associated with protein supplementation. We believe that this addition not only strengthens the scientific rigor of our manuscript but also provides a comprehensive understanding of the complex relationship between whey protein intake and type 2 diabetes risk. We hope that this modification adequately addresses your concerns and enhances the overall quality of our paper.

Figure 1 needs to be clarified in resolution

Thank you for your observation. We have added a version of the figure with a better resolution.

I recommend that you add a paragraph explaining the relationship between athlete health and whey protein in the discussion section, because athletes are among the group of people who use whey protein most effectively in the world. 

We appreciate your recommendation to include a detailed analysis of the relationship between athlete health and whey protein consumption. In response, we have added a comprehensive paragraph in the Discussion section ( Line 289-301)  highlighting the benefits of whey protein for athletes, its role in muscle recovery and performance enhancement, as well as the importance of its correct usage. We have also incorporated pertinent studies to support these claims. This addition aims to provide a clearer understanding of why athletes are among the most effective users of whey protein worldwide and underscores the necessity of personalized nutrition strategies for this population. We believe this addition significantly strengthens our manuscript and hope it satisfactorily addresses your suggestion

Reviewer 2 Report

Comments and Suggestions for Authors

The review carried out by the authors is very interesting, since they show the pros and cons of whey protein consumption. Regarding the search in the databases, I consider them adequate, as well as the inclusion and exclusion criteria. I found it interesting that preclinical and clinical trials were included to give more strength to their research, and with this they are in a neutral point, that is, they are neither for nor against the use of this whey protein, since in some cases as they describe it they favor its use and in other cases they do not. Finally, I consider the aspects that address renal and hepatic functioning, acne, microbiota, behavioral and emotional aspects, as well as bone metabolism and other health effects, to be relevant.

Detailed comments can be found in the attachment.

Author Response

Thank you for your encouraging feedback on our review of whey protein consumption. We are pleased that the comprehensiveness of our research, particularly the inclusion of both preclinical and clinical trials, resonated well with you. Our aim was to present an unbiased and holistic view of whey protein consumption, highlighting both its benefits and drawbacks in various contexts. We appreciate your recognition of our efforts to maintain a neutral stance, neither advocating for nor against whey protein use, but rather presenting the evidence as it stands. Additionally, your acknowledgment of the relevance of our discussion on renal and hepatic function, dermatological effects like acne, impacts on the microbiota, behavioral and emotional considerations, bone metabolism, and other health effects is highly valued. We believe that these aspects are crucial in understanding the multifaceted nature of whey protein's impact on health. Your feedback reinforces our commitment to delivering well-rounded and scientifically sound research.

Please find the detailed response in the attachment.

Reviewer 3 Report

Comments and Suggestions for Authors

Why wasn't a systematic review done to get stronger evidence? Suggestion: Systematic review with identifying PICO.

I found a similar systematic review: “Vasconcelos QDJS, Bachur TPR, Aragão GF. Whey protein supplementation and its potentially adverse effects on health: a systematic review. Appl Physiol Nutr Metab. 2021 Jan;46(1):27-33. doi: 10.1139/apnm-2020-0370. Epub 2020 Jul 23. PMID: 32702243. Justify the reasons for the similar review, please.

The abstract is not complete and comprehensive. It has additional words, while some expressions such as “depending on usage patterns” are not clear.

Introduction: Explanations of Tables 1 and 2 should be integrated in the text.

Materials and Methods: The latest version of flow diagram (2020) should be used.

References: Most of the references are old and up-to-date ones are less used.

Comments on the Quality of English Language

Minor editing of English language required.

Author Response

Dear Editors and Reviewers,

first, we would like to thank you for the valuable impulses that allowed us to improve the quality of the manuscript. All changes made are highlighted by yellow color, in the revised version of the manuscript, to facilitate the review process.

Hoping that we have satisfied your requests as much as possible, we kindly ask you to re-evaluate our paper. 

The Authors

REVIEWER N.3

Why wasn't a systematic review done to get stronger evidence? Suggestion: Systematic review with identifying PICO.

Thank you for your insightful query regarding the choice of our review methodology. The decision to conduct a narrative rather than a systematic review was driven by several considerations. Firstly, our objective was to provide a broad, comprehensive overview of the diverse effects of whey protein across various health aspects, which extends beyond the scope typically covered in a systematic review. We aimed to encapsulate a wide range of studies, encompassing varying methodologies and contexts, to present a holistic picture of the current understanding in this field. Furthermore, the heterogeneity in the study designs, populations, and outcomes of interest in whey protein research poses a challenge for the stringent inclusion and exclusion criteria of a systematic review. This diversity, while enriching the field with varied insights, makes it challenging to synthesize data in the systematic review format without losing the nuances and breadth of information available. However, we acknowledge the value of a systematic review in providing strong evidence through a focused and structured analysis based on the PICO framework. In light of your suggestion, we recognize that a systematic review could complement our current work by offering a more targeted analysis of specific aspects of whey protein consumption and its effects. Such an approach could be an excellent avenue for future research, allowing for a detailed exploration of specific questions within this domain. In conclusion, while our review takes a broader, narrative approach to encompass a wide spectrum of research, we appreciate the merit in conducting a systematic review and consider it a valuable recommendation for future research endeavors in this field.

I found a similar systematic review: “Vasconcelos QDJS, Bachur TPR, Aragão GF. Whey protein supplementation and its potentially adverse effects on health: a systematic review. Appl Physiol Nutr Metab. 2021 Jan;46(1):27-33. doi: 10.1139/apnm-2020-0370. Epub 2020 Jul 23. PMID: 32702243. Justify the reasons for the similar review, please.

We thank the reviewer for pointing this paper out. Of course we are aware of the presence of this well-conducted and comprehensive systematic review published by Vasconcelos et al. in 2020. Our aim was mainly an update of these already published information that we accounted for, although our manuscript has a different outline: specifically we have extended Vasconcelos’s findings almost doubling the number of studies included (21 vs 11 studies) for both preclinical (5 vs 9)  and clinical trials (6 to 12). Due to the “exclusion” criteria we imposed, being a review, we did not cite Vasconcelos in the references, but we screened all the studies included in their systematic review and selected in our work those eligible for our criteria.

Moreover, in the discussion (paragraph 4.6 & 4.7) we addressed new issues that Vasconcelos et al. did not mention:  possible allergy events and Vitamin K Deficiency issues for infant formula rich in WP; WP byproducts and conjugates that can be generated before and after ingestion. Although these fields do not have enough evidence to be included in our review process, we believe that our work represents an increase in knowledge and an update of literature after the well-written and detailed systematic review provided by Valconcelos et al.

The abstract is not complete and comprehensive. It has additional words, while some expressions such as “depending on usage patterns” are not clear.

Thank you for the suggestion, we have indeed modified the sentence to be clearer and extended the abstract information with : “ in relation to different posologies in a variety of settings. Our study suggests caution for the protein intake in situation of hepatic and renal compromised functions, as well as in acne susceptibility, while possible beneficial effects can be achieved for the intestinal microbiota, humoral and behavioural level, and finally bone and muscle mass in elderly. “

Introduction: Explanations of Tables 1 and 2 should be integrated in the text.

We appreciate your suggestion to integrate explanations of Tables 1 and 2 into the introduction of our manuscript. We have now included a detailed description of these tables in the introduction, providing context and clarity on the types and compositions of whey protein supplements.  

Materials and Methods: The latest version of flow diagram (2020) should be used.

Thank you very much for your suggestion regarding the use of the PRISMA 2020 flow diagram. We would like to clarify that our review methodology is based on the Scale for the Assessment of Narrative Review Articles (SANRA) guidelines. The review was conducted using multiple databases including PubMed, Scopus, and Google Scholar. Given the nature of our review as a narrative synthesis, we have not utilized the PRISMA guidelines, which are typically applied to systematic reviews. We appreciate your attention to detail and your commitment to ensuring the quality and rigor of our work. We hope this clarification aligns with the standards of the Journal and serves to enhance the understanding of our methodology for the readers. We have added a version of the previous figure (FIG 1) with a better resolution for readers’ convenience.

References: Most of the references are old and up-to-date ones are less used.

Thank you for your valuable feedback concerning the recency of the references in our manuscript. We have taken your comment into consideration and have updated our reference list to include more current studies. In our revised manuscript, we now incorporate a balanced mix of foundational research and recent studies, ensuring that our review is both grounded in established knowledge and informed by the latest developments in the field. This update enriches the relevance and timeliness of our work, aligning with the evolving nature of research on whey protein and its health implications. We appreciate your guidance in enhancing the quality of our manuscript.

Reviewer 4 Report

Comments and Suggestions for Authors

There are some issues that require attention.

1. The resolution of Figure 1 is too low to discern details.

2. Please also review the information provided on the Healthcare template and update all references to match the style of the Journal of Healthcare. Line-by-line suggested edits are recommended.

3. What are your conclusions for each section in Section 3 (Results)? For example, what are your findings regarding liver function, kidney function, acne, gut function and microbiota, emotional and behavioral influences, and bone metabolism? Provide comments on each, and what is the takeaway message?

Author Response

Dear Editors and Reviewers,

first, we would like to thank you for the valuable impulses that allowed us to improve the quality of the manuscript. All changes made are highlighted by yellow color, in the revised version of the manuscript, to facilitate the review process.

Hoping that we have satisfied your requests as much as possible, we kindly ask you to re-evaluate our paper. 

The Authors

REVIEWER N.4

  1. The resolution of Figure 1 is too low to discern details.

Thank you for your observation. We have added a version of the figure with a better resolution.

  1. Please also review the information provided on the Healthcare template and update all references to match the style of the Journal of Healthcare. Line-by-line suggested edits are recommended.

Thank you for pointing out the necessity of aligning the references with the Journal of Healthcare's style guidelines. We understand the importance of adhering to the journal's formatting requirements. In this regard, we would like to highlight that the manuscript will undergo a thorough proofreading process provided by MDPI. Their proofreading service is known for its efficiency and attention to detail, particularly in ensuring that references are formatted correctly in accordance with the journal's style guidelines. We trust that this process will address any inconsistencies and ensure that the final version of the manuscript fully complies with the Journal of Healthcare's formatting standards.

  1. What are your conclusions for each section in Section 3 (Results)? For example, what are your findings regarding liver function, kidney function, acne, gut function and microbiota, emotional and behavioral influences, and bone metabolism? Provide comments on each, and what is the takeaway message?

Thank you for your suggestion to provide clear conclusions for each section in our results. To effectively communicate the findings, we have compiled a synthesis table (Table 5) at the end of our manuscript. This table presents a summarized view of the key findings and conclusions on the impact of whey protein on various health aspects including liver and kidney function, acne, gut health, emotional and behavioral influences, and bone metabolism. We believe that this table not only addresses your request but also enhances the readability and clarity of our manuscript, making the key takeaway messages more accessible to the readers. In addition, upon suggestion of another reviewer we also have included an overview of the findings in the abstract. We thank you for your valuable suggestion that enhances the readability of the manuscript and helps the reader with retaining the takeaway messages.

Round 2

Reviewer 3 Report

Comments and Suggestions for Authors

-

Comments on the Quality of English Language

-